# Effect of Autochthonous Nepalese Fruits on Nutrient Degradation, Fermentation Kinetics, Total Gas Production, and Methane Production in *In-Vitro* Rumen Fermentation

**DOI:** 10.3390/ani12172199

**Published:** 2022-08-26

**Authors:** Rajan Dhakal, Manuel Gonzalez Ronquillo, Einar Vargas-Bello-Pérez, Hanne Helene Hansen

**Affiliations:** 1Department of Veterinary and Animal Sciences, Faculty of Health and Medical Sciences, University of Copenhagen, Grønnegårdsvej 3, DK-1870 Frederiksberg C, Denmark; 2Departamento de Nutrición Animal, Facultad de Medicina Veterinaria y Zootecnia, Instituto Literario 100, Universidad Autónoma del Estado de México, Toluca 50000, Estado de México, Mexico; 3Policy and Development, Department of Animal Sciences, School of Agriculture, University of Reading, P.O. Box 237, Earley Gate, Reading RG6 6EU, UK

**Keywords:** fruits, rumen fermentation, methane, fatty acids, environment

## Abstract

**Simple Summary:**

Autochthonous fruits with naturally active chemicals such as polyphenols and tannins have been used in animal feeds to cure animal diseases due to their beneficial effects. This study aims to determine the effect of autochthonous Nepalese fruits on nutrient degradation, fermentation kinetics, total gas production, and methane production in *in*-*vitro* rumen fermentation. We found that the addition of Nepalese autochthonous fruits increases total fatty acid concentration and decreases methane production with increasing doses.

**Abstract:**

The objective of this study was to determine the effect of autochthonous Nepalese fruits on nutrient degradation, fermentation kinetics, total gas production, and methane production in *in-vitro* rumen fermentation. The fruits of *Terminalia* *chebula* (HA), *Terminalia* *bellirica* (BA), and Triphala churna (TC), a commercial mixture with equal parts (33.3% DM basis) of *Phyllanthus* *emblica*, *Terminalia* *bellirica*, and *Terminalia* *chebula*, were used. These were tested at three inclusion levels of 20% 40% and 100% of the total sample (as dry matter) in maize silage (MS). MS was used as a control (0% additive). These 10 treatments were tested for two 48-h incubations with quadruplicate samples using rumen fluid from 2 heifers. Total gas production (TGP: mL at standard temperature and pressure (STP)/g DM), methane production (expressed as % and mL/g DM), and volatile fatty acids were determined. After incubations, the filtrate was used to measure pH and volatile fatty acids (VFA), while the residue was used to measure degraded dry matter (dDM) and calculate the partitioning factor (PF48) and theoretical short-chain fatty acid concentration (tVFA). Rumen fluid pH linearly (*p* < 0.01) decreased in all treatments with increasing dose during fermentation. The CH_4_% was less in all three treatments with 100% autochthonous plants than in control, but there were no significant linear or quadratic effects for increasing BA, HA, and TC doses. The PF48 increased for all treatments with a significant linear and quadratic effect (*p* < 0.05) of increasing dose. Compared to MS, the inclusion of autochthonous plants increased the total volatile fatty acids, with no significant dose effects. The tVFA linearly decreased (*p* > 0.05) with an increasing dose of BA and HA. All treatments showed quadratic effects on tVFA (*p* < 0.05) with increasing dose. Increasing TC dose linearly (*p* < 0.05) and quadratically (*p* < 0.05) increased total VFA, while increasing HA dose had only a quadratic (*p* < 0.05) effect on total VFA. All treatments reduced total gas production (TGP) and methane concentration (CH_4_%) when compared to MS. The tested autochthonous fruits can be used as additives with a basal feed diet to reduce enteric methane emissions. The most effective anti-methanogenic treatment was 40% HA, which resulted in 18% methane reduction.

## 1. Introduction

Globally, the livestock sector contributes significantly to livelihoods, economic growth, and the use of natural resources [1]. Ruminant production has a valuable role in sustainable agriculture systems [2]. Ruminant microorganisms can ferment and degrade structural carbohydrates in forage cell walls, providing energy and nutrients to the host animal [3]. The rumen microbial ecosystem consists of diverse populations of bacteria, archaea, fungi, and protozoa. These microorganisms break down fibers by hydrolysis to produce fatty acids (VFA) and microbial protein and, at the same time, hydrogen (H_2_) and carbon dioxide (CO_2_) are the primary initial products [4]. This fermentation process contributes around 75% of the energy to the ruminant. During fermentation, the produced H_2_ in the rumen is utilized by methanogens to produce methane (CH_4_) as a byproduct. The eructated methane is a greenhouse gas (GHG) with a global warming potential 25 times higher than that of carbon dioxide (CO_2_) [5].

Strategies for methane reduction from ruminants are an important global concern to diminish global climate change. Methane produced during rumen fermentation contributes to a loss of feed efficiency: between an estimated 2 and 12% of gross energy intake [6]. Therefore, reducing enteric methane emissions from ruminants has economic, environmental, and nutritional benefits. Various strategies have been investigated to reduce this methane, including the use of direct-fed microbes [7,8,9,10], algae [11,12,13], plant extracts [14,15,16], natural and synthetic chemicals [17,18,19], and exogenous enzymes [20,21,22]. Many of these strategies have shown a promising effect on the mitigation of enteric methane emissions.

In subtropical and tropical climate regions of the Indian subcontinent (Bangladesh, Bhutan, India, Nepal, Pakistan, and Sri Lanka), *Terminalia bellirica* (BA), *Terminalia chebula* (HA), and the commercial product Triphala Churna (TC) are commonly used in Ayurveda medicine. Humans use the fruits of these plants as a laxative [23], and in animals, they are used to treat diarrhea and prevent protozoa growth [24]. *Phyllanthus emblica* contains the antibacterial compound phyllemblin and flavonoids such as quercetin, kaempferol, rutin, quercetin 3-β-d-glucopyranoside, and kaempferol 3-β-d-glucopyranoside [25,26]. Triphala Churna (TC) contains phenolic acid and corilagin along with various other compounds and has antibacterial bacterial and anti-protozoal properties [27,28]. *Terminalia bellirica* contains different polyphenols including gallic acid, chebulic acids, and corilagin [23]. Methanol extract of BA and HA was found to contain 133 and 127 phenols (mg/mL gallic acid equivalent per 100 mg plant extract) and 138 and 219 flavonoids (mg/mL quercetin equivalent per 100 mg plant extract), respectively [29]. Levels of total polyphenols found in *Terminalia chebula* and *Terminalia bellirica* fruits were 125.9 and 223.7 g/kg DM. The polyphenols are mainly hydrolyzable tannins and ellagitannins with an additional 34 different types of polyphenols [23]. Triphala Churna has exhibited *in-vitro* antimicrobial properties comparable to ampicillin and also contains antioxidant properties [30]. Hydrolyzable tannins such as simple gallate esters, ellagic acid derivatives, glycosides, ellagitannins, and essential oils can manipulate the rumen fermentation [31].

According to Peterson et al. [32], the polyphenols in TC can modulate the human gut microbiome, promote beneficial *Bifidobacteria* spp. and *Lactobacillus* spp., and inhibit the growth of undesirable gut microbes. Likewise, the lactic acid producing bacteria (LAB), lactic acid utilizing bacteria (LUB), or other species such as *Bifidobacterium* spp. or *Enterococcus* spp. improve dry matter intake, milk yield, fat corrected milk yield, and milk fat content [33].

Recent reviews have identified a promising anti-methanogenic potential of autochthonous fruits and plant extracts. Therefore, subtropical and tropical climate regions can use local natural resources as additives for enteric methane mitigation in ruminants [16,34,35]. Thus, the objective of this *in-vitro* study was to determine the dose-response effects of selected Nepalese fruits as feed additives on fermentation kinetics, CH_4_ and VFA production, and dry matter degradability. The aim was investigated by determining a dose–response effect of these plants on CH_4_ production and dry matter degradation, volatile fatty acid production (VFA), and estimation of energy utilization *in-vitro*, which is the first step in determining the usefulness of these plants in livestock production systems. We hypothesized that *Terminalia chebula*, *Terminalia bellirica* and Triphala Churana added to a diet based on maize silage would significantly reduce enteric methane without adverse effects on rumen fermentation parameters and nutrient degradation.

## 2. Materials and Methods

### 2.1. Treatments and Donor Animals

*Terminalia bellericia* (BA), *Terminalia chebula* (HA), and Triphala Churna (TC), which is an equal mixture of *Phyllanthus emblica*, *Terminalia bellericia*, and *Terminalia chebula* fruits (33% of DM of each), were tested at four different doses (0%, 20%, 40%, and 100% DM) using maize silage (MS) as the basal diet for the additives (Table 1). The amount of additive in each treatment was chosen to represent the maximum and minimum possibility of additive use, according to local custom.

Sun-dried fruits and the commercial product (TC) were bought from a retail herbal shop in Kathmandu, Nepal in 2017, and all samples of fruits were ground in a cyclone mill using a 2 mm sieve. A 500 mg sample for each treatment was weighed into a 100 mL Duran^®^ bottle, tested during a 48-h fermentation with quadruplicates of each treatment, and replicated. Each bottle was fitted with an automated pressure sensor system (ANKOM^RF^ Technology, Macedon, NY, USA; pressure range: from −69 to +3447 kPa; resolution: 0.27 kPa; accuracy: ±0.1% of measured value). Each module sends measurements via a receiving base station to an attached computer. Bottles with rumen fluid but without feed (BL) were included to determine baseline fermentation. For each fermentation, rumen fluid (solid and semi-solid phases) was collected from two rumen cannulated heifers at the Large Animal Hospital from the University of Copenhagen (Taastrup, Denmark). The heifers were fasted for 12 h before rumen fluid collection. The cannulated animal use was authorized by Danish law (research animal license no. 2012-15-2934-00648). Animals were fed ad libitum haylage (containing 85% of dry matter, 7.5 MJ/kg metabolizable energy, and 11% crude protein) for more than six weeks before the experiment.

### 2.2. In-Vitro Fermentation

A four-part buffer solution was prepared as described by Menke and Steingass [36]. The buffer media was flushed with CO_2_ for two hours to ensure anaerobic conditions, and the temperature of the media was maintained at 39 °C before the addition of rumen fluid. A reduction agent of sodium sulfide and sodium hydroxide was added approximately 10 min before the addition of the rumen fluid. An equal amount of rumen fluid from each heifer was filtered through a double layer of washed commercial cheesecloth before adding to the buffered media. The rumen fluid was added to the buffer in a 1:2 ratio. Ninety mL of the rumen–buffer inoculum was added to each bottle, flushed with CO_2_, and closed with the ANKOM^RF^ module. A gas tight (SKC, Flex Foil PLUS) evacuated sample bag was attached to the vent valve tube of the module to collect all produced gas. During the fermentation, there was a live recording interval of 60 s, a recording interval of 10 min, and a release pressure of 0.75 PSI. The modules were incubated in a ThermoShaker (Gerhardt Königswinter, Germany) at 39 °C with 40 rotations per minute. After the end of the experiment, the bottles were placed on ice to stop fermentation. Subsequently, the pH (HECH pH31^®^) of each bottle was measured. After that, the fluid from each bottle was filtered using vacuum suction to collect undigested residue in a pre-weighed filter bag with a porosity of 25 µ (Ankom F57).

### 2.3. Dry Matter and Ash Analysis

The final dry matter content of the pure samples was determined by drying the samples at 105 °C for 8 h. The residual weight determined ash content after burning the samples overnight at 520 °C in a muffle furnace.

### 2.4. Volatile Fatty Acid Determination

Volatile fatty acid determination was completed as previously reported by Vargas-Bello-Pérez [37]. To measure volatile fatty acids (VFA), 5 mL of filtered rumen fluid was collected and placed into a 7 mL test tube previously filled with a one mL metaphosphoric solution (5:1 ratio) for VFA determination. This mixture was homogenized using a vortex (IKA^®^ MS 3 Basic, Wilmington, NC, USA) for one minute and stored at −20 °C until analysis. After that, the samples were incubated for thirty minutes at room temperature. Before analysis, the samples were thawed at 5 °C and centrifuged at 14,000 rpm for 15 min. The supernatant was transferred to 2.5 mL Eppendorf tubes and centrifuged at 14,000 rpm for 10 min. The supernatant of this second centrifuged fluid was filtered through a syringe filter with 0.2 µm pore diameter (MiniSart Syringe Filter, Satorius, Goettingen, Germany), and a 1 µL filtered sample was subsequently kept in 2 mL GC vials for further analysis.

The VFA content and profile were determined by gas chromatography (Nexis GC-2030, Shimadzu Scientific Instruments Inc., Kyoto, Japan) equipped with a 30 m wall-coated open-tubular fused-silica capillary column (Stabilwax-DA; 30 m × 0.32 mm i.d., 0.25 μm film thickness; Shimadzu, Riverwood Drive, Columbia, USA). The running time per sample was 8.71 min. The oven temperature was programmed for 145 °C for 3 min and then increased from 145 °C to 245 °C at 16.6 °C/min. The injector and flame ionization detector were kept at 250 °C. Gas flows were 24, 32, and 200 mL/min for N_2_, H_2_, and synthetic air. A standard curve was made using a volatile fatty acid mix from Sigma Aldrich (St. Louis, MO, USA).

### 2.5. Methane Determination

The methane content in the gas-tight bags was measured directly after the 48-h incubation in a gas chromatograph (GC) (Agilent 7820A GC, Agilent Technologies, Santa Clara, CA, USA). The GC was equipped with a HPPLOT Q column (30 m × 0.53 mm × 40 µmm), and H_2_ was used as the carrier gas. The column flow was 5 mL/min and the TCD detector was set to 250 °C with a reference and makeup flow of 10 mL/min. A 250 µL gas sample was taken from each gasbag after the contents were mixed and manually injected into the GC and replicated samples from each gasbag. Run time was 3 min at an isothermal oven temperature of 50 °C. Calibration curves were calculated from standards containing 1%, 2.5%, 5%, 10%, 15%, and 25% CH_4_ in nitrogen (Mikrolab A/S, Aarhus, Denmark). The total methane concentration (% of collected gas) produced was thereafter calculated.

### 2.6. Fiber Analysis

After filtration of the fermented feed, the filter bags were air-dried at room temperature for 24 h and then dried at 100 °C for 2 h. Fiber analyses were undertaken on incubated and non-incubated samples in the ANKOM200 fiber analyzer for NDF and ADF [38,39]. Heat stable alpha amylase and sodium sulfite were used according to ANKOM protocol [39] but with amylase addition in the wash and, first, an NDF rinse cycle for incubated samples.

### 2.7. Calculations and Statistical Analysis

The pressure, dry matter residue, and methane concentration from the blank treatments were used to correct gas production, dry matter degradation, and methane production. After that, results from blanks were not used, and only results of blank corrected variables are shown.

Percent dry matter, NDF (dNDF), and ADF (dADF) degradation was calculated as follows. All values are on a dry matter basis:

(DM, NDF, or ADF content in the unfermented sample − (DM, NDF, or ADF residue after fermentation)/DM, NDF, or ADF in the unfermented sample) × 100

The Ideal Gas law was used to convert cumulative pressure to mL gas at standard temperature and pressure per gram of dry matter:PV = n R T
where P = Pressure in PSI, V = Volume of gas (mL), n = moles of gas, R = gas constant, and T = Temperature in Celsius (39.5 °C) at STP. Finally, the mL of gas produced per gram dry matter was calculated using the following formula:mL gas/g DM = V/ DM in sample. 

The partitioning factor at 48 h (PF_48_) and theoretical short-chain fatty acid concentration (tVFA) were calculated according to Elghandour et al. [40]. The PF_48_ calculates fermentation efficiency as the ratio of DM degradability *in-vitro* (dDM, mg) to the volume (mL) of GP at 48 h. Theoretical short chain fatty acid concentration (tVFA) [40] was calculated as:tVFA(mmol/200 mg DM) =0.0222 GP − 0.00425 
where GP is the 24 h net gas production (mL/200 mg DM at STP).

### 2.8. Statistical Analyses

The response to each treatment was analyzed separately. For all responses (TGP, dDM, dNDF, dADF, PF_48_, CH_4_%, tVFA, and VFA), the following linear mixed model was used to compare the means of the treatments.
Y_ijk_ = µ + D_i_+ R_j_ + E_ijk_
where Y_ijk_ is the observation of the ith dose of treatment (D_i_), µ is the general mean, and R_j_ is the random effect of fermentation (two incubations). The R function lme [41] for mixed models investigated differences in digestible dry matter and the extracted curve parameters. Orthogonal contrasts were used to test the linear and quadratic trends. Statistical analyses were performed in R version 3.5.1 (https://www.R-project.org/, accessed on 8 August 2020), and the ‘drc’ package [42] was used for curve fitting. The response curves for each treatment were tested using a sigmoidal or exponential curve, and the model with the least AIC (Akaike information criterion) was chosen to generate predicted parameter values.

#### 2.8.1. “Groot” Model

This model is described in Groot et al. [43].
$$Y_{t}=A/(1+(B^{C}/t^{C}))$$
where *Y_t_* is the total gas yield at time *t* (mL/g DM incubated), *A* is the asymptotic gas production (mL/g DM), *B* is the time at which half the asymptotic amount of gas has been produced (in min), and *C* is the constant determining the steepness of the curve. 

#### 2.8.2. Exponential Model with No Intercept (or Intercept = 0)

This model corresponds to a simple exponential model.
$$Y_{t}=b(1-e^{-ct)})$$

*Y_t_* is the total gas produced at time *t* (in mL/g DM incubated), *b* is the asymptotic gas production (mL/g DM), and *c* is the constant determining the curves′ steepness. 

## 3. Results

The autochthonous Nepalese fruits had more NDF but less ADF than MS, suggesting more hemicellulose and pectin contents (Table 2). In particular, *Terminalia bellirica* is primarily made up of hemicellulose, with less than 5% cellulose and lignin in the DM. In general, CP contents of the pure autochthonous fruits were lower than MS, and TC had a greater polyphenols content than BA and HA.

Increasing doses of BA showed both a linear (*p* < 0.05) and quadratic (*p* < 0.05) relationship with H1, Vmax, Tmax, and gas production at 24 h (Table 3). However, A1 and gas production at 9 h and 48 h had a quadratic relationship (*p* < 0.05) with dose, but only for BA. Increasing doses of HA linearly (*p* < 0.05) decreased Vmax. However, only a quadratic effect (*p* < 0.05) of the dose was seen for A1, Tmax, and gas production at 48 h. Both a linear (*p* < 0.05) and quadratic effect (*p* < 0.05) were seen for H1 and gas production at 24 h. Increasing doses of TC linearly (*p* < 0.05) decreased H1 and Vmax. A quadratic effect (*p* < 0.05) was observed for A1, Tmax, and gas production at 6, 9, 24, and 48 h.

Dry matter degradation showed both a quadratic (*p* < 0.05) and linear relationship (*p* < 0.05) for BA and HA, with increasing dose, but this increase was only significant as a linear function for TC (Table 4). BA and HA doses have a linear (*p* < 0.05) and quadratic effect (*p* < 0.05) on dNDF, and TC only has a quadratic (*p* < 0.05) effect of dose on dNDF. BA has a linear (*p* < 0.05) and HA has a quadratic (*p* < 0.05) effect of dose on dADF. The percent of the original sample ADF that was degraded was greater than the percent original sample NDF that was degraded for BA, while the original sample ADF was less degraded than NDF for HA, TC, and maize silage. Both pH and PF48 were linearly related to dose for BA, HA, and TC, whereas PF48 was additionally related quadratically (*p* < 0.05). The increase in the dose of BA and HA was related both linearly (*p* < 0.05) and quadratically (*p* < 0.05) to tVFA.

TC had linear (*p* < 0.05) and quadratic (*p* < 0.05) effects of dose on total VFA, and HA had only a quadratic (*p* < 0.05) effect of dose on total VFA (Table 5). The BA dose had a linear (*p* < 0.05) effect on iso-butyric acid and a quadratic effect (*p* < 0.05) on valeric acid. The HA dose had a quadratic (*p* < 0.05) effect on caproic acid, and the TC dose had a quadratic effect (*p* < 0.05) on acetic and propionic acid

## 4. Discussion

### 4.1. Nutrient Degradation and Total Gas Production

The chemical composition of feed is an essential factor for predicting the true digestibility of dry matter or organic matter in *in- vitro* gas production research [44]. In theory, the greater the DM degradability, the greater the gas production [45].

Hemicellulose and pectin are considered to be more digestible than cellulose [44]. BA, HA, and TC had less cellulose and more hemicellulose contents than MS. However, the increased degradation of ADF compared to NDF in BA suggests that the hemicellulose fraction decreased the overall NDF degradation. Additionally, the least overall dDM of BA suggests that using HA and TC plants for dairy cow feed in subtropics and tropics could be more beneficial than BA. Amylolytic microbes will degrade the hemicellulose and pectin rapidly while the cellulolytic microbes begin to break down the cellulose. However, the content of these heterogeneous groups of polysaccharides vary between different fruit species, plant part, and geographical location [46]. Fermentation efficiency (PF48) of MS with all the Nepalese fruits linearly increased with increased doses. This measure of fermentation efficiency is calculated by and is based on the basic assumption that greater degradation produces more gas but does not consider microbial biomass growth.

However, this basic assumption is contradicted by the greater degradability yet decreased TGP of 100% HA and TC compared to MS. This was unexpected because both TC and HA had relatively greater contents of hemicellulose and pectin, less cellulose, less lignin, and a greater amount of dDM than a pure basal diet (MS). Therefore, they would be expected to have a greater degradation than MS but also a greater gas production. However, Patra et al. [47] reported a significantly greater *in-vitro* gas production when using concentrated extract of *A. cocinna*, *E. officinalis*, and *T. bellirica* in buffalo rumen fluid. This could be expected, as the extract would not contain the structural carbohydrates. It is important to note that ADF contents in HA and TC and maize silage are more than double that in BA. The linear decease in degradation of ADF (dADF) with increasing doses of BA may support the suggestion of the specific BA hemicelluloses retarding the overall degradation.

A decreased TGP was seen in the present study with increasing BA, HA, and TC doses. However, these decreases were not linear. Digestible dry matter (dDM) decreased linearly with an increasing additive dose (Table 3). Non-linear effects were also seen by Jayanegara et al. [48], who tested the effects of mixing two or more different forages and found that the results of CH_4_ production, CH_4_ to VFA ratio, and CH_4_ to TGP ratio were not linear. They found that forages with high levels of phenols exhibited non-additive effects, as the phenols interact with proteins and decrease digestibility [39]. The Nepalese fruits used in this research could be expected to have phenolic levels that would affect the linearity of response.

### 4.2. Methane Production and Volatile Fatty Acids

The chemical composition of the feed affects the rumen microbiome which, in turn, plays a vital role in total gas production, CH_4_ production, and VFA concentration [46]. Changing the chemical composition of the feed through additives is one of the strategies to affect fermentation. Methane production is influenced by additives or supplements, including probiotics in the feed [49,50,51]. We found a numerical decrease in methane concentration (CH_4_% of collected gas) from pure autochthonous fruits compared to MS, but a significant linear reduction was not found with increasing dose of autochthonous fruits. Patra et al. [47] investigated the effects of methanol extracts of the same plant (HA: *T. chebula*) on *in-vitro* reduction in methane. Methane was reduced by more than 95% when adding 0.83% and 1.67% extract to a basal 1:1 forage to concentrate ratio. Patra et al. [52] also reported reduced methane emissions per kilogram of digested DM intake in sheep fed 1% HA (DM) in a 1:1 concentrate to forage ratio. In yet another study, the inclusion of 2% and 4% HA (DM) showed significant inhibition of methane production *in-vitro* using buffalo rumen fluid and, in the same research, when using the levels tested *in-vitro* in in vivo feeding trials, a non-significant reduction in CH_4_ of up to 17% was observed [35]. The same trend was found, though not significant, in the present study. The most effective anti-methanogenic treatment was 40% HA among the treatments with additives, which resulted in an 18% methane production reduction.

The methane production decreased with increasing doses of autochthonous fruits without a difference in VFA’s total production or profile. Total gas, total VFA, and methane production (%), and digestibility are closely related [53,54]. The inclusion of the additives to MS increased total VFA production by between 4 and 9%, suggesting that the inclusion of the Nepalese autochthonous fruits increased the nutritional value of MS. The tVFA decreased for both the pure HA, BA, and TC and when the fruits were used as an additive to MS. This discrepancy is because the tVFA was calculated from the gas produced at 24 h, at which time fermentation is not complete, while the measured VFAs were measured after 48 h of fermentation. During fermentation, acetate and butyrate production are the main sources of gas production [55,56,57,58]. The HA, BA, and TC additive dose increase from 0 to 40% decreased the acetic acid proportion. A decrease in the proportion of acetic demands an increase in other VFAs. An increase in propionic acid would cause a reduction in TGP and CH_4_ due to the stoichiometry of VFA production [55]. While not significantly different in the present research, an increase in propionic acid could be the reason for reducing the CH_4_ proportion or yield and reduced TGP. The acetic to propionic acid ratio also did not differ significantly between MS and the different dose of additives with MS. However, the total gas produced at 48 h by MS was greater than MS with the different doses of autochthonous Nepalese fruits and a significant quadratic response was observed, indicating that the dose had passed the vertex of the parabola. Pure HA produced less and pure BA more propionic acid due to the greater proportion of hemicellulose [44] and pectin in BA than HA, explaining the measured decrease in TGP and CH4. This supports the conclusion that autochthonous plants can reduce methane without negative consequences on rumen fermentation.

The lack of VFA profile differences may be due to hydrolyzable tannins that interact with MS protein, reducing fermentability and, thereby, TGP and dDM. Aderao et al. [59] found that using polyphenol rich plant leaves, such as *Acacia nilotica* and *Ziziphus nummularia*, in complete feed blocks reduced methane emission. Likewise, the inclusion of combined hydrolyzable tannins and condensed tannins at a concentration of 1.5% dietary DM decreased CH_4_ emission without having negative effects on performance [60]. The finding from these studies is similar to our results, in that the tested fruits are known to contain polyphenols, tannins, and bioactive compounds [47]. Gallic acids are phenolic compounds found within hydrolyzable tannins and unbound to other compounds. An increased level of gallic acid has been found to increase total short chain fatty acid production [61]. Gallic acid is known to be found in the Nepalese autochthonous fruits, and this may explain the increased total VFA in the treatments with MS mixed with these plants.

However, it is important to consider the anti-nutritional properties of hydrolyzable tannins and account for the negative influence when planning rations [39]. Therefore, the judicious use of agro-industrial by-products naturally rich in polyphenols, when included in ruminant diets, can reduce the feeding cost for farmers, conferring benefits to dairy products in terms of improving the quality and sustainability of their production [62].

### 4.3. Limitations of the Study and Future Perspectives

It is important to note that the Nepalese fruits depend on local and seasonal availability and may be produced and procured by small or medium-size entrepreneurs, resulting in fluctuating availability. Our results create a benchmark for the use of Nepalese fruits for anti-methanogenic effects at different doses. However, further research should be performed using in vivo studies to analyze performance parameters, milk composition, and blood metabolites.

## 5. Conclusions

Overall, this study showed that the addition of Nepalese autochthonous fruits increases volatile fatty acid concentrations but did not decrease methane production significantly. A numerical trend of decreasing methane concentration was seen when adding all fruits from 40 to 100% to maize silage. The hemicellulose content in BA was reduced overall in NDF degradation, unlike that in HA, TC, or maize silage. A more detailed study needs to be conducted to confirm these findings. This study identified that commonly used fruits for cattle feed in Nepal may reduce methane in subtropical and tropical climate regions.

## Figures and Tables

**Table 1 animals-12-02199-t001:** Description of Nepalese fruit-based treatments.

Name of the Treatments	Dose	Composition of the Treatment
Maize silage (MS)	0	100% Maize silage (MS)
*Terminalia bellirica* (BA)	20	20% *Terminalia bellirica* + Maize silage (MS)
	40	40% *Terminalia bellirica* + Maize silage (MS)
	100	100% *Terminalia bellirica*
*Terminalia chebula* (HA)	20	20% *Terminalia chebula* + Maize silage (MS)
	40	40% *Terminalia chebula* + Maize silage (MS)
	100	100% *Terminalia chebula*
Triphala Churna (TC)	20	20% Triphala Churna + Maize silage (MS)
	40	40% Triphala Churna + Maize silage (MS)
	100	100% Triphala Churna

**Table 2 animals-12-02199-t002:** Chemical composition of the Nepalese fruits and maize silage (g/100 g of dry matter).

Analysis	Maize Silage	*Terminalia bellirica*	*Terminalia chebula*	Triphala Churna
Dry matter	93.1	92.7	92.8	94.9
Neutral detergent fiber	44.2	91.5	77.3	80.6
Acid detergent fiber	24.1	3.7	6.8	8.1
Acid detergent lignin	2.1	0.7	0.8	1.4
Ash	4.3	2.8	2.7	4.4
Crude protein	8.5	4.7	3.0	3.9
Polyphenols	-	127.6 *	133 *	158.7 *

* Phenolic content expressed as mg/mL gallic acid equivalent per 100 mg plant extract [29].

**Table 3 animals-12-02199-t003:** Gas production kinetics parameters and *in-vitro* gas production (IVGP) parameters for Nepalese fruits (BA, HA, TC) at 0%, 20%, and 40% inclusion in MS.

GP Parameters					IVGP Parameters				
Dose	A1 (mL Gas STP/g DM)	H1 (h)	Vmax (mL Gas STP/g DM Per min)	Tmax (min)	6 hTGP/g DM	9 hTGP/g DM	12 h	24 h	48 h
*Terminalia bellirica* (BA)									
0	194.89	16.92	1.73	741.25	7.09	30.39	51.86	150.66	186.16
20	185.74	15.77	1.29	663.75	9.27	32.65	49.17	128.46	168.13
40	172.12	15.38	1.29	545	9.33	38.88	57.27	124.67	155.49
100	116.28	8.50	1.18	348.66	13.25	53.64	70.76	105.94	113.49
Linear	0.46	<0.001	<0.001	<0.001	0.91	0.66	0.76	0.01	0.15
Quadratic	<0.001	0.01	<0.001	<0.001	0.13	0.003	0.69	<0.001	<0.001
SEM	3.68	0.57	0.13	37.73	1.67	1.54	2.81	6.31	5.87
*Terminalia chebula* (HA)									
0	194.89	16.92	1.73	741.25	7.09	30.39	51.86	150.66	186.16
20	176.35	16.12	1.20	565	9.45	35.62	58.99	121.16	158.28
40	156.10	15.81	1.07	521.25	10.49	35.93	54.12	107.90	137.89
100	71.92	4.52	4.01	50.5	25.25	50.97	56.32	70.51	70.07
Linear	0.71	<0.001	0.02	0.7	0.29	0.63	0.85	0.007	0.10
Quadratic	<0.001	<0.001	0.40	<0.001	0.02	0.01	0.13	<0.001	<0.001
SEM	5.53	0.42	0.51	42.4	1.98	3.22	3.40	5.50	6.42
Triphala Churna (TC)									
0	194.89	16.92	1.73	741.25	7.09	30.39	51.86	150.66	186.16
20	196.89	15.77	1.27	636.25	9.11	35.60	53.84	129.94	172.45
40	177.37	14.97	1.24	531.25	11.24	41.56	62.77	127.66	161.83
100	127.53	9.28a	2.81	97.5	24.05	62.21	73.8	106.22	121.97
Linear	0.38	0.03	0.03	0.91	0.52	0.86	0.39	0.13	0.58
Quadratic	0.05	0.95	0.74	<0.001	0.03	0.004	0.34	<0.001	<0.001
SEM	4.45	0.49	0.37	34.84	1.93	2.98	3.41	6.77	6.52

A1: fitted curve maximum gas production mL/g DM, H1: time at which half of A1 is produced, Vmax: maximum rate of gas production, Tmax: time at maximum gas production, and SEM: pooled standard error of the mean.

**Table 4 animals-12-02199-t004:** Dose responses for pH, dry matter, neutral detergent fiber and acid detergent fiber degradation (dDM, dNDF, dADF), PF48, tVFA, and CH_4_% after fermentation of three Nepalese fruits (BA, HA, TC) at 0%, 20%, and 40% inclusion in MS.

Dose.	pH	dDM^1^%	dNDF^1^%	dADF^1^%	PF48h	tVFA (mmol)	CH_4_%
*Terminalia bellirica* (BA)							
0	6.94	76.2	36.6	32.1	2.21	3.73	8.15
20	6.83	69.8	41.2	44.6	2.69	3.3	10.05
40	6.84	65.8	39.2	47.1	3.05	3.2	8.03
100	6.90	69.3	24.5	40.0	2.95	2.5	4.31
Linear	0.03	<0.001	0.001	0.001	0.03	0.03	0.19
Quadratic	0.13	<0.001	0.001	0.27	0.001	<0.001	0.55
SEM	0.07	0.69	1.33	2.51	0.31	0.10	1.24
*Terminalia chebula* (HA)							
0	6.94	76.2	36.8	32.1	2.21	3.73	8.15
20	6.84	69.1	43.6	35.0	2.77	3.05	8.78
40	6.85	69.5	36.7	24.6	3.54	2.72	7.83
100	6.69	84.1	13.7	9.8	5.84	1.55	1.76
Linear	0.01	<0.001	<0.001	0.72	0.05	<0.001	0.16
Quadratic	0.24	0.03	<0.001	0.01	0.002	<0.001	0.32
SEM	0.08	0.45	1.36	1.84	0.29	0.10	1.44
Triphala Churna (TC)							
0	6.94	76.2	36.8	32.1	2.21	3.73	8.15
20	6.85	72.5	40.8	31.9	2.57	3.34	9.11
40	6.84	70.6	36.2	23.9	2.87	3.23	8.03
100	6.90	79.2	20.7	13.9	3.34	2.54	4.58
Linear	0.02	<0.001	0.10	0.57	0.02	0.23	0.43
Quadratic	0.25	0.99	0.05	0.07	<0.001	<0.001	0.68
SEM	0.08	0.41	2.44	2.74	0.27	0.14	1.18

**Table 5 animals-12-02199-t005:** Volatile fatty acid production and composition after 48 h of *in-vitro* rumen fermentation of Nepalese fruits (BA, HA, TC) at 0%, 20%, and 40% inclusion in MS.

Dose	Total VFA (mmol/L)	Acetic Acid (mol/100 mol)	Propionic Acid (mol/100 mol)	Iso-Butyric Acid (mol/100 mol)	Butyric Acid (mol/100 mol)	Isovaleric Acid (mol/100 mol)	Valeric Acid (mol/100 mol)	Caproic Acid (mol/100 mol)
*Terminalia bellirica* (BA)								
0	66.7	42.1	24.0	2.21	22.4	4.67	2.85	1.69
20	63.6	39.1	25.1	1.78	25.1	3.68	2.85	2.27
40	63.1	42	24.0	1.75	24.3	3.61	2.98	1.84
100	61.9	42.1	25.5	1.83	21.8	3.53	2.80	2.12
Linear	0.74	0.67	0.75	0.09	0.16	0.14	0.54	0.65
Quadratic	0.71	0.42	0.16	0.33	0.95	0.23	0.047	0.115
SEM	6.90	1.80	1.02	0.18	1.40	0.45	0.97	0.3
*Terminalia chebula* (HA)								
0	66.7	42.1	24.0	2.21	22.4	4.66	2.85	1.69
20	68.3	42.7	26.4	1.74	21.1	3.56	2.95	1.44
40	61.9	42.9	24.4	1.86	22.4	3.86	3.00	1.56
100	46.9	46.5	19.5	2.04	24.8	4.00	2.40	0.75
Linear	0.53	0.88	0.21	0.12	0.67	0.25	0.35	0.73
Quadratic	0.015	0.27	0.34	0.40	0.79	0.12	0.28	0.03
SEM	2.89	2.03	1.45	0.19	2.17	0.43	0.24	0.24
Triphala Churna (TC)								
0	66.7	42.1	24	2.21	22.4	4.67	2.85	1.69
20	67.2	40.9	24.5	1.85	24.4	3.90	2.86	1.66
40	67.9	42	23.9	1.67	24.6	3.47	2.72	1.57
100	58.6	50	20.2	1.73	20.6	3.32	2.82	1.23
Linear	0.03	0.18	0.14	0.11	0.11	0.18	0.21	0.96
Quadratic	0.003	0.04	0.01	0.44	0.45	0.34	0.36	0.42
SEM	1.16	1.44	0.53	0.19	1.3	0.46	0.06	0.22

SEM: Pooled standard error of the mean.

## Data Availability

The data presented in this study are available on request from the corresponding author.

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
