# Peer review of "Effect of Autochthonous Nepalese Fruits on Nutrient Degradation, Fermentation Kinetics, Total Gas Production, and Methane Production in In-Vitro Rumen Fermentation"

_animals, 2022, doi:10.3390/ani12172199_

Round 1
Reviewer 1 Report
1. Line 34-25 Do you mean 20%, 40%, and 100% ? pls double check and modify this sentence to be consistent with the material part.
2. Line 29 "SCFAs" The abbreviation should be defined upon first use
3. The NDF content of the fruits > 92% and ADF content ≤8.1% ? This result could be unreliable.
4. Table 4, the dADF% was greater than dNDF%, why ?
5. The conclusion section should be revised since the data was insufficient to support that 40% inclusion of Terminalia chebula (HA) was the potential anti-methanogenic.
Author Response
"Please see the attachment."

Reviewer 2 Report
Dear Authors, the manuscript is very interesting, the studies carried out on Terminalia chebula (HA), Terminila belliricia (BA) and Triphala churna (TC), must however report some clarifications: 1- were other gases detected besides the production of methane gas? 2- The method of analysis is gas chromatography, however the work does not report any gas chromatogram; The two points mentioned must be adequately described.Best regards
Author Response
"Please see the attachment."

Reviewer 3 Report
The aim of the presented research was to investigate the potential of three different plants, originating from Nepal, to reduce methane formation in ruminants using in vitro methods. The topic of the research is relevant and interesting as for farmers in tropical and subtropical regions where investigated plants are easily available but also for academia, since the authors clearly indicated the possible future research regarding investigated fruit material.
The layout of the paper is sufficiently good. the methodology is clear and very detailed. The results are presented in a manner so that potential readers can clearly follow given discussion. The discussion is very well supported by prior investigations. The English is sufficiently good and some small corrections are needed. I would recommend the authors to read in detail the manuscript once more and to correct some minor spelling format errors. I would also recommend to the authors to exclude the sentence from line 71, since the same details on Triphala Churna are given later in section 2.1 of the manuscript.
Overall, the given conclusions are supported by the given results and would recommend this paper for publishing after minor revisions.
Author Response
"Please see the attachment."

Round 2
Reviewer 1 Report
accept